**Data Availability Statement:** All relevant data are within the paper and its Supporting Information files.

**Funding:** This study is supported by a grant from the Departmental General Research Fund, funded

# Changes in corneal astigmatism and near heterophoria after smartphone use while walking and sitting

**Tsz Wing Leung**[1,2]* , **Chui-Ting Chan**[1], **Chi-Hin Lam**[1], **Yuk-Kwan Tong**[1], **Chea-Su Kee**[1]

1 School of Optometry, The Hong Kong Polytechnic University, Hong Kong SAR, China, 2 The Centre for Myopia Research, School of Optometry, The Hong Kong Polytechnic University, Hong Kong SAR, China

☯ These authors contributed equally to this work.
* jeffrey.TW.leung@polyu.edu.hk

## Abstract

### Background/Aims

Smartphone use has become an indispensable part of our daily life. The handy design and powerful processor allow smartphone users to perform diversified tasks even when walking. This study aimed to investigate and compare the optical aftereffect and vergence adaptation of using a smartphone while walking and sitting.

### Methods

Twenty-nine young healthy adults (aged 19 to 24 years old) with normal binocular and accommodative functions were recruited. Participants were asked to watch a movie for 30 minutes using a smartphone while either walking on a treadmill or sitting on a chair. Corneal aberrations and near heterophoria were measured before and after smartphone use by a corneal topographer and modified Thorington heterophoria test, respectively.

### Results

Using the smartphone while walking induced a change in corneal H/V astigmatism, becoming 0.11±0.03 μm less negative (two-way ANOVA repeated measures, Bonferroni post-hoc test, p = 0.001). This optical aftereffect was significantly higher than after smartphone use while sitting by 0.10±0.03 μm (paired t-test, p = 0.003). Although smartphone use did not result in a significant change in near heterophoria (Bonferroni post-hoc test, p > 0.15), the vergence adaptation showed relatively more eso- or less exo-deviation by 0.79±0.36$^{\Delta}$ in the walking than the sitting condition (paired t-test, p = 0.037).

### Conclusions

Eyecare practitioners should be cautious of the potential optical after effect and vergence adaptation after prolonged smartphone usage.

by the Hong Kong Polytechnic University, Hong
Kong (Grant number: P0031874).

**Competing interests:** The authors have declared
that no competing interests exist.

## Introduction

In this digital era, using a smartphone has become an indispensable part of our daily life. The
number of smartphone users is growing dramatically worldwide, reaching 3.5 billion users by
2020 [1]. The powerful processor and well-designed mobile apps allow diversified functions.
Users can enjoy video streaming, play games, learn from diverse resources, engage in social
media communication, and perform many other activities all with a single device. Inevitably,
these robust features induce users to spend more time on their mobile devices. For example, in
Hong Kong, where over 90% of adults own at least one smartphone [2], average use of mobile
devices is 2.4 hours each day, with 20% of users spending more than 4 hours per day [3]. Unlike
traditional computers that can only be used at a fixed working station, the design of smartphones
allows owners to use their mobile devices almost everywhere, even while they are walking.

Although smartphones are hugely productive tools that bring great convenience to our
daily life, using a smartphone while walking can pose considerable dangers. According to a
recent survey conducted by the American Academy of Orthopaedic Surgeons [4], about one-
third of the smartphone users frequently use their smartphones while walking for non-speech
activities (such as reading emails/websites, texting, playing games, or taking selfies). Of these
users, 26% had encountered walking incidents, ranging from bumping into obstacles without
injuries to sprains or fractures. Previous studies have shown that using a mobile device while
walking alters gait kinematics [5], provokes risky street-crossing behaviors [6], and reduces the
awareness of roadside surroundings [7]. Despite this variety of hazards associated with smart-
phone use, how sitting and walking affects our vision during smartphone use remains elusive.

The overuse of smartphones substantially increases the risk for digital eye strain (or com-
puter vision syndrome), which has become a recognized global health problem [8, 9]. A recent
survey of over 10,000 US adults revealed that 65% of respondents experienced symptoms
related to the use of digital devices [10]. Digital eye strain can be categorized into internal and
external symptoms [11]. Internal symptoms, including blurred vision, diplopia, eyestrain, and
headache, are linked to the stress on the refractive and binocular vision systems of the eye,
whilst external symptoms, including excessive tearing, dryness, burning, and irritation, are
closely related to the dry eye syndrome. While these symptoms are usually transient, they can
frequently cause a significant visual disturbance to individuals.

Because of the reduced screen dimensions and small font size, a smartphone is usually used
at a short viewing distance, on average 36.3 cm for texting and 32.2 cm for internet viewing
[12], posing substantial stress to the optical and fusional vergence systems of the eye. For
instance, prolonged near-work, using either video display terminals or hard-copy reading
materials, leads to a transient increase in both against-the-rule corneal astigmatism (negative
cylindrical axis at 90 degrees) [13–15] and myopia [16, 17], which could temporally degrade
both distance and near vision. Short-term adaptation of functional vergence to near tasks also
shifts near heterophoria [18–21], with the magnitude of deviation associated with the subjec-
tive report of visual fatigue [19]. These transient changes in optical and vergence systems of
the eye can vary with the gaze position. Collins et al. [15] and Buehren et al. [14] reported that
near tasks that required a more downward gaze induced more against-the-rule corneal astig-
matism, probably due to the increased eyelid pressure exerted onto the cornea. In addition, the
resting position of tonic vergence varies with the inclination angle of the eye, as a downward
gaze usually results in a near shift of heterophoria [22, 23]. Raap and Ebenholtz [24] also found
that eye movement could prevent adaptation of the fusional vergence system to convergent sti-
muli (i.e., the stimulation of inward movement of both eyes, such as near tasks). It is worth
noting that contrary to sitting, walking involves continuous movement of the gaze position to
coordinate with head and body motions [25]. Thus, this study hypothesized that using

smartphones while sitting and walking might place different stress on the visual systems and affect the optics and fusional vergence of the eye.

In this study, we investigated the effect of smartphone use on optical qualities and binocular functions of the eye. Specifically, we measured changes in corneal astigmatism and near heterophoria after 30 minutes of smartphone use between two experimental conditions: while walking on a treadmill or sitting on a chair. This study provides additional evidence on the potential ocular hazards of smartphone usage under different viewing conditions.

## Materials and methods

Twenty-nine young, healthy Chinese adults (age: 18–24 years old) with normal binocular and accommodative functions participated in this study. All participants had spherical-equivalent refractive errors of +4.00DS to -6.00DS and astigmatism of 1.50DC or less, with corrected visual acuity at distance better than logMAR 0 in either eye. None had more than 1D of anisometropia. They were all free from strabismus, amblyopia, and ocular disease, and had no history of ocular surgeries. This study complied with the Declaration of Helsinki and was approved by the Human Subjects Ethics Committee of the Hong Kong Polytechnic University (HSEARS20150715001). Written informed consent was obtained from each eligible participant after explaining the experimental procedures.

Before starting the study, all participants underwent an eye examination to determine their refractive status and ocular health. Non-cycloplegic subjective refraction, using maximum-plus-maximum-acuity as the endpoint [26], was conducted to measure their refractive errors. Participants were required to wear their habitual spectacle corrections when either principal power meridian was more than 0.75 D. If the corrective power of their spectacle lenses deviated more than 0.50 D from their subjective refraction, they were advised to obtain a new pair of spectacles. Otherwise, they were excluded from this study. None of the participants was a regular contact lens wearer. A list of standard clinical procedures was also conducted to rule out binocular vision and ocular accommodation anomalies: cover test for distance and near dissociated heterophoria, Royal Air Force rule for near point of convergence and amplitude of accommodation, ±2D flipper for accommodative facility, and Risley prism on phoropter for distance and near fusional vergence reserve. Binocular and accommodative functions which exceeded 95% confidence intervals (CI) of the age norm were considered as abnormal [27]. Eligible participants were invited to complete the following procedures.

### Experimental protocols

This study investigated the effect of using a smartphone on the corneal optical quality and binocular vision while either walking or sitting and was conducted over two separate visits. The order of walking and sitting conditions was assigned randomly. Each visit started at about the same time of the day (± 2 hours) to avoid potential diurnal variation of corneal biometry [28]. Before and after the smartphone use in each visit, participants first underwent the near dissociated heterophoria test, followed by corneal aberrometry measurement (see detailed procedures below). These procedures took less than five minutes. The near heterophoria was determined before the corneal aberrometry because the aberrometer blocks one eye during the measurement, which could have interfered with the vergence system following smartphone use.

At each visit, participants were first seated in a dark room for three minutes to dissipate any transient changes in accommodation and vergence before starting the measurements [29, 30]. The experiment began with baseline measurements of near heterophoria and corneal aberrometry. The participants, wearing any habitual spectacles, used a smartphone (LG G3 Stylus D690N, Republic of Korea) to watch a Korean variety show called "Running Man", a popular

foreign TV program in Hong Kong, for 30 minutes with Chinese subtitle (subtitle's size: ~ 3mm). None of the participants knew Korean, and reading the subtitle was necessary. The screen (size: 5.5 inches; weight 163 g; resolution: 540 x 960 pixels) was held horizontally to maximize the display dimension, with the luminance set to its maximum level (~ 570 cd/m$^2$ for a white background). The ambient room lighting was maintained at about 300 lux. Participants were either sitting on a height-adjustable office chair or walking on a treadmill (Model: Cadence 4.9 Weslo, West Logan, UT) while using the smartphone. The speed of the treadmill was set at 0.7 m/s with an inclination angle of 10 degrees. For safety reasons, a console key was attached to participants' clothes. In an emergency, the console key would be detached to slow down and stop the walking belt, although this never occurred during the study. An examiner stood nearby to monitor the whole process. Immediately after 30 minutes of smartphone use, near heterophoria and corneal aberration measurements were repeated.

## Corneal aberration measurement

Corneal topography was measured by an i.Profiler aberrometer (Carl Zeiss Vision, German), which incorporates a Placido-disk based videokeratoscope (number of rings: 22; up to 3,425 measuring points). Once the instrument was aligned with the pupil center, it automatically captured five consecutive corneal topographies and averaged the wavefront data to reduce any random measurement errors. The whole process took about 1–2 minutes. The corneal wavefront aberrations were derived from a 5-mm pupil diameter using Zernike polynomials up to the 7th radial order. The lower-order corneal aberrations, i.e., oblique astigmatism and H/V astigmatism, were averaged for data analysis. Higher-order corneal aberrations were excluded from the data analysis because they contributed to less than 10% of the total corneal aberration and were likely to have minimal clinical implications on visual quality. This study only determined corneal, but not ocular aberrations, for two reasons. Firstly, the study measured aberrations under a natural pupil without pharmacological dilation, and thus, thirteen participants (i.e., about 45% of our sample) could not achieve a 5 mm or larger pupil diameter. Secondly, because aberrometry was performed without cycloplegia, the aberrometer might have stimulated proximal accommodation and affected the internal (or lenticular) and overall aberrations of the eye. Nevertheless, because ocular astigmatism mostly originates from the cornea [31, 32], and the near-work induced change in ocular aberrations was highly correlated with the changes in corneal aberrations [14], the measurement of corneal astigmatism should, at least in part, reflect the changes in the overall optical quality of the eye after smartphone use.

To aid in the interpretation of the corneal H/V astigmatism, Z(2, 2), and oblique astigmatism, Z(2, -2), we also presented corneal astigmatism in diopter units by converting the Zernike coefficients ($C_2^{+2}$ and $C_2^{-2}$) to power vectors (J0 and J45) [33].

$$J0 = \frac{-C_2^{+2} 2\sqrt{6}}{r^2}$$

$$J45 = \frac{-C_2^{-2} 2\sqrt{6}}{r^2}$$

where r is the pupil radius. The J0 and J45 are the astigmatic components with horizontal/vertical and oblique axes, respectively, and can be converted to the cylindrical form (Cyl) for a correcting lens as follow [34].

$$Cyl = 2\sqrt{J0^2 + J45^2}$$

**Table 1. Demographic and baseline information.**

|  | Mean (95% confidence intervals) |
|---|---|
| **Age** | 21.5 y (22.03, 20.97) |
| **Gender** | 40% female |
| **Spherical Equivalent** | -2.67 DS (-3.50, -1.84) |
| **Astigmatism** | 0.57 DC (0.38, 0.76) |
| **Oblique Astigmatism Z(2, -2)** | 0.090 μm (0.043, 0.137) |
| **H/V Astigmatism Z(2, 2)** | -0.86 μm (-0.76, -0.95) |
| **Near Heterophoria** | 4.76$^\Delta$ Exophoria (4.01, 5.50) |

## Near heterophoria measurement

The modified Thorington method was used to measure near dissociated heterophoria because the procedure is fast and straightforward with good inter-examiner repeatability [35–38]. During the measurement, a Maddox rod with the striations oriented horizontally was placed in front of the right eye to create a vertical striated light. Participants were asked to fixate at a light spot located at the center of a Bernell Muscle Imbalance Measure (MIM) card [35] at 40 cm and report the lateral displacement of the vertical striated light. They were reminded to always keep the scales of the MIM card clear throughout the whole process. The reported scales represent the near dissociated heterophoria.

# Results

## Demographic and baseline information

Participants' demographic information, baseline corneal aberrations, and near heterophoria are summarized in Table 1. All participants had a negative corneal H/V astigmatism with an average of (mean±SE) -0.86±0.09 μm, equivalent to about +0.67 D of J0 astigmatism [33]. The magnitude of corneal oblique astigmatism was relatively low (0.09±0.05 μm, equivalent to about -0.07 D of J45 astigmatism), about 9.5 times smaller than the corneal H/V astigmatism. The majority of our participants exhibited exophoria (n = 26, 5.46±0.71$^\Delta$ exophoria), one had orthophoria, and two had esophoria (1.5$^\Delta$ and 2.5$^\Delta$ esophoria, respectively). Table 2 summarizes the changes in corneal astigmatism and near heterophoria before and after smartphone use.

**Table 2. Summary of changes in corneal astigmatism and near heterophoria after 30-minute smartphone use while walking and sitting.**

|  |  | Walking | Sitting |
|---|---|---|---|
| **Oblique Astigmatism Z(2, -2) (μm)** | *Pre* | 0.070±0.045 | 0.110±0.052 |
|  | *Post* | 0.058±0.048 | 0.073±0.049 |
| **H/V Astigmatism Z(2, 2) (μm)** | *Pre* | -0.883±0.097* | -0.831±0.092 |
|  | *Post* | -0.769±0.095* | -0.816±0.096 |
| **Near Heterophoria ($^\Delta$)** | *Pre* | 4.72±0.77 | 4.79±0.77 |
|  | *Post* | 4.37±0.71† | 5.24±0.73† |

Two-way ANOVA repeated measures was used to examine the effect of time and experimental conditions on corneal astigmatism and near heterophoria. There were statistically significant interactions between the effects of time and experimental conditions on H/V astigmatism ($F(1, 28) = 8.09$, $p = 0.008$) and near heterophoria ($F(1, 29) = 4.78$, $p = 0.037$).
Bonferroni post-hoc test: Pre vs Post * $p = 0.001$; Walking vs Sitting † $p = 0.009$.

## Changes in corneal astigmatism

The results showed that smartphone use while walking for only 30 minutes significantly increased corneal H/V astigmatism, but was not significant for use while sitting (Fig 1, two-way ANOVA repeated measures: experimental conditions X time interaction: $F(1, 28) = 8.09$, $p = 0.008$). Compared with the baseline, the corneal H/V astigmatism after walking using the

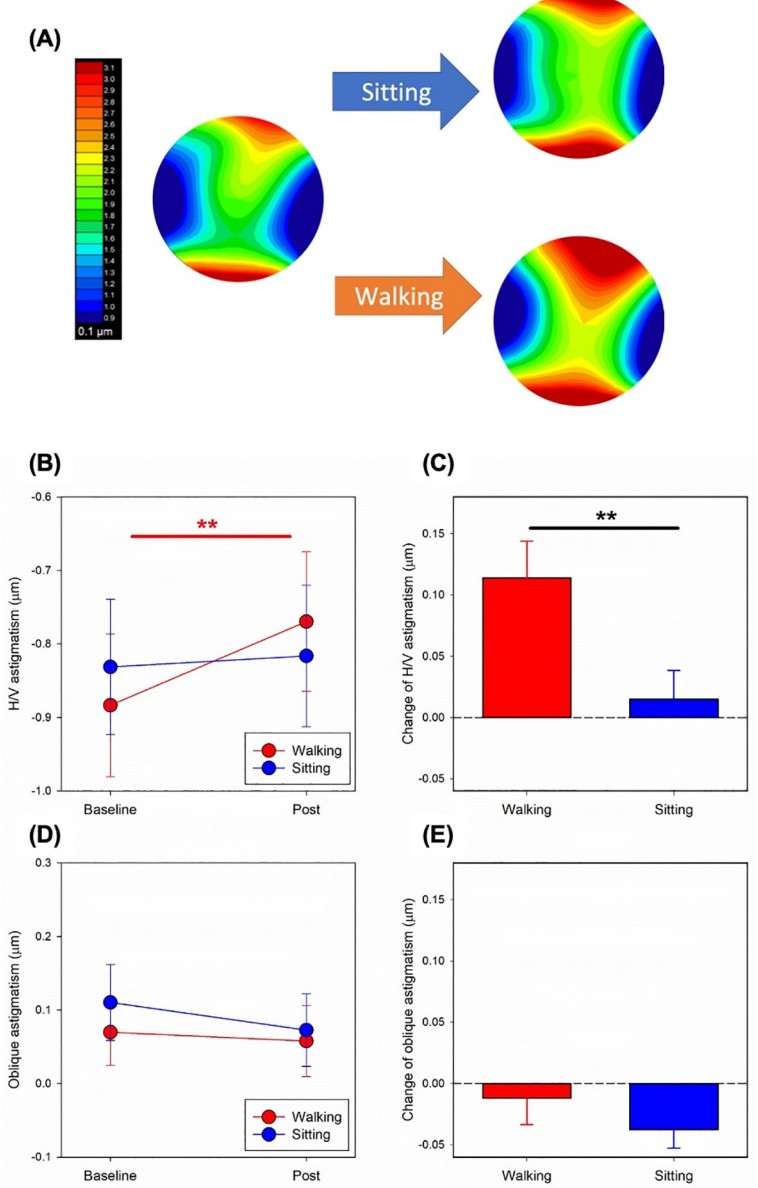

**Fig 1. Optical aftereffects following 30 minutes of smartphone use. A)** Results for subject S01: The change in corneal wavefront was more obvious after smartphone use while walking than sitting. **B) Corneal H/V astigmatism:** The corneal H/V astigmatism became significantly less negative after smartphone use while walking, but no significant change was found while sitting. **C) Corneal H/V astigmatism:** The change in H/V astigmatism was significantly more positive while walking than sitting. **D & E) Corneal oblique astigmatism:** The optical aftereffect for the corneal oblique astigmatism was not statistically significant, and no significant difference in the change in oblique astigmatism was found between groups. *Red symbols*: Walking; *Blue symbols*: Sitting. Bonferroni post hoc test or paired t-test: ** *p < 0.01*.

smartphone for 30 minutes became less negative by 0.11±0.03 μm (Fig 1B, Bonferroni post-hoc test, $t = 3.80$, $p = 0.001$). However, in the sitting condition, the change in corneal H/V astigmatism between the baseline and post-smartphone use was not statistically significant ($t = 0.65$, $p = 0.53$).

Comparison of the two experimental conditions showed that corneal H/V astigmatism at the baseline was slightly more negative (0.05±0.03 μm) in the walking than the sitting condition (Fig 1B, Bonferroni post-hoc test, $t = 1.86$, $p = 0.074$), but after using the smartphone for 30 minutes, corneal H/V astigmatism was 0.05±0.02 μm more positive in the walking than the sitting condition ($t = 2.04$, $p = 0.051$). However, both comparisons failed to reach statistical significance.

To better understand the difference between the two experimental conditions, change in corneal H/V astigmatism before and after smartphone use (i.e., post–baseline) was compared and was found to be significantly more positive while walking than sitting (Fig 1C, paired t-test, $t = 3.31$, $p = 0.003$).

The effect of smartphone use on the corneal oblique astigmatism was not statistically significant, neither during walking nor sitting (Fig 1D, experimental conditions: $F_{(1, 28)} = 4.06$, $p = 0.053$; time: $F_{(1, 28)} = 3.42$, $p = 0.075$; interaction: $F_{(1, 28)} = 0.98$, $p = 0.330$). The change in corneal oblique astigmatism (i.e., post–baseline) was also not significantly different between the two experimental conditions (Fig 1E, paired t-test, $t = 1.77$, $p = 0.09$).

### Changes in near heterophoria

Near heterophoria showed a different pattern of vergence adaptation between the two experimental conditions (Fig 2A, two-way ANOVA repeated measures: experimental condition X

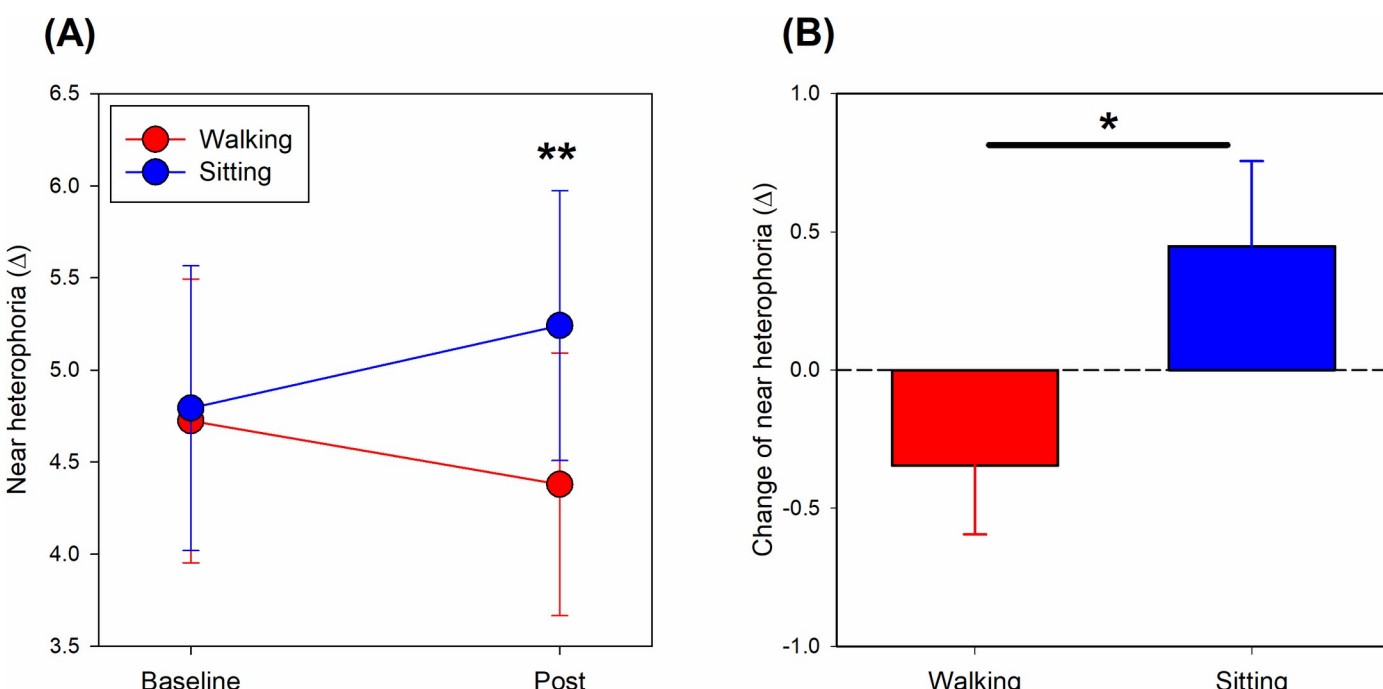

**Fig 2. Vergence adaptation following 30 minutes of smartphone use. A)** In the post-smartphone use session, near heterophoria showed significantly less exo-deviation (i.e., less positive) after walking than sitting, while no significant difference was found at the baseline. **B)** Change in near heterophoria showed significantly less exo-deviation (or more eso-deviation) for walking than sitting. Positive: Exo-deviation; negative: Eso-deviation. *Red symbols*: Walking; *Blue symbols*: Sitting. Bonferroni post hoc test: $^{**}$ $p < 0.01$, paired t-test: $^{*}$ $p < 0.05$.

time interaction: $F(1, 29) = 4.78$, $p = 0.037$). While near heterophoria at the baseline was comparable between the two experimental conditions (Bonferroni post-hoc test, $t = 0.17$, $p = 0.863$), the near heterophoria after 30 minutes of smartphone use showed $0.86 \pm 0.30^\Delta$ less exo-deviation (i.e., the eyes tended to turn out less) while walking than sitting ($t = 2.83$, $p = 0.009$). Comparison of the baseline and post-smartphone use sessions revealed that the eye tended to show more exo-deviation while sitting and less exo-deviation while walking after using the smartphone. However, none of these changes reached statistical significance (both $t < 1.38$, $p > 0.15$).

Changes in near-heterophoria (i.e., post–baseline) between the two experimental conditions were also compared. Smartphone use while walking resulted in the eyes tending to shift to more eso- or less exo-deviation than when sitting (Fig 2B, paired t-test, $t = 2.19$, $p = 0.037$).

## Discussion

With improvements in technology, the smartphone has become more portable and powerful, enabling users to perform diversified tasks in different environments. This study compared the optical aftereffect and vergence adaptation of using a smartphone while walking and sitting. The findings revealed that using a smartphone while walking resulted in a less negative corneal H/V astigmatism, and this change was significantly higher than that observed while sitting. The near heterophoria after walking with smartphone use was also more esophoric (or less exophoric) than sitting.

This study confirms that intensive near work reduced with-the-rule or induced against-the-rule corneal astigmatism [13–15]. When performing near tasks, the palpebral aperture usually becomes narrower, increasing the pressure the eyelid exerts onto the cornea and alters corneal toricity [13]. The magnitude of corneal astigmatic changes appears to be associated with the gaze position: a downward gaze, such as reading and performing microscopy, induces more against-the-rule astigmatism than a forward gaze, such as using computers [15]. It was observed that participants tended to have a more downward gaze when using smartphones while walking than when sitting. The change in gaze direction is probably an involuntary reflex to ensure safety by increasing the visual field of the walking path. This change to a downward gaze may explain the difference in the optical aftereffects between walking and sitting while using a smartphone (Fig 1A). However, other factors, such as the extraocular muscle forces generated during convergence [39], inter-blinking duration [40], and magnitude of accommodation [41], could also contribute to the variation of corneal shape and aberrations after smartphone uses.

The average change in corneal H/V astigmatism after walking use was equivalent to about 0.18 DC against-the-rule astigmatism. In this study, eleven participants showed >0.25 DC of induced against-the-rule astigmatism (H/V astigmatism > 0.16 µm), while two had >0.50 DC changes (H/V astigmatism > 0.32 µm). Although this study did not directly measure ocular aberrations, it has been shown that the near-work induced change in corneal aberrations was highly correlated with the change in ocular aberrations [14]. Thus, the small but significant corneal astigmatism induced after smartphone use might degrade the overall optical quality of the eyes, and also distance and near vision. When compared to previous studies (data were converted from Zernike coefficients to cylindrical error to aid in interpretation), the induced against-the-rule corneal astigmatism after smartphone use while walking appears to be higher than that after reading (~0.04 DC to 0.09 DC [13–15]) or using a computer (~0.03 DC [15]), but less than that after using a microscope (~0.21 DC [10]). Although the average astigmatic change was relatively small, it is possible that the experimental protocol may have led to an underestimation of optical aftereffects. It is worth noting that corneal aberration was measured

after the near heterophoria assessment, i.e., the refractive measurement was not performed immediately after the smartphone use. The induced against-the-rule astigmatism could have decayed during the heterophoria assessment because participants were required to look straight ahead at the MIM chart, which may have led to the eyelid pressure exerted onto the cornea to have been reduced. The decay in optical aftereffects may also explain why the change in corneal astigmatism in the sitting position was not statistically significant, even though using a smartphone while sitting also involved a downward gaze.

Prolonged or intensive near work can also impose stress on the oculomotor system. After 30 minutes of smartphone use, the change in near heterophoria displayed relatively less exo- or more eso-deviation following walking than sitting, suggesting that the two experimental conditions imposed different stresses on the eyes. However, comparison of near heterophoria between baseline and post-smartphone use, showed there was no significant difference for either sitting or walking positions. Indeed, the effects of near work on binocular vision are inconsistent between studies. Whilst some studies revealed an esophoric shift following intensive near-work [19–21], others either showed an exophoric shift [18, 42, 43] or no significant change in heterophoria [44]. The discrepancies between studies could arise from the differences in their methodologies, including the near-work duration (from 20 to 90 mins), subject recruitment criteria (from children to adults), methods of vergence assessment, working distance (6.5 cm to 33 cm), and difficulty of the near task (reading, watching cartoon, visual search, and using virtual reality). This study indicated a difference in vergence adaptation between using smartphones in the walking and sitting conditions, although the change in near heterophoria did not reach clinical significance (limit of agreement for the modified Thorington test: $2.3^\Delta$) [37]. Further studies are needed to better understand whether and how the direction of vergence adaptation relates to the oculomotor system.

This study allowed participants to use the smartphone freely, without restriction on their posture and viewing distance, to simulate a natural viewing condition. The change in corneal astigmatism and near heterophoria reported in this study should help better understand the visual disturbance suffered by smartphone users and the potential hazards for using the smartphone while walking. However, because this study did not control the viewing conditions and measure head and eye positions, the reasons behind the changes in corneal astigmatism and heterophoria are not conclusive. The coordination of the head and eye movement during the smartphone use (e.g., whether the user moved the head instead of the eye to stabilize the gaze) could affect the dynamic of eyelid pressure exerted on the cornea and the stress acting on the fusional vergence system. While it is not the primary purpose of this study to investigate the underlying causes of digital eye strain, it is worthwhile for further studies to be performed on how working distances, gaze direction, walking gait, and dynamic visual environment impose stress on the eyes.

To conclude, smartphone use while walking induced against-the-rule corneal astigmatism (i.e., a less negative corneal H/V astigmatism) and shifted the near heterophoria to be less exophoric when compared with the sitting condition. These significant optical and binocular vision changes–although small in magnitude–were observed in our young participants with normal vision. It would be important to investigate whether patients with binocular vision and accommodative anomalies would suffer from similar or even more effects.

## Supporting information

**S1 File. Data file.** The study raw data.
(XLSX)

## Author Contributions

**Conceptualization:** Chea-Su Kee.

**Data curation:** Chui-Ting Chan, Chi-Hin Lam, Yuk-Kwan Tong.

**Formal analysis:** Tsz Wing Leung, Chui-Ting Chan.

**Investigation:** Chui-Ting Chan, Chi-Hin Lam, Yuk-Kwan Tong.

**Project administration:** Chui-Ting Chan, Chi-Hin Lam, Yuk-Kwan Tong.

**Supervision:** Tsz Wing Leung, Chea-Su Kee.

**Writing – original draft:** Tsz Wing Leung, Chui-Ting Chan.

**Writing – review & editing:** Tsz Wing Leung, Chea-Su Kee.

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
