## [Decision Letter · Decision Letter 0]

17 Jun 2020

PONE-D-20-16428

Smartphone Use while Walking Induced Optical and Vergence Aftereffects

PLOS ONE

Dear Dr. Leung,

Thank you for submitting your manuscript to PLOS ONE. After careful consideration, we feel that it has merit but does not fully meet PLOS ONE’s publication criteria as it currently stands. Therefore, we invite you to submit a revised version of the manuscript that addresses the points raised during the review process.

Although the work presented is relevant and timely, I agree with both reviewers who consider major revisions to be required prior to publication. Please revise the manuscript with particular attention to the rationale, ensuring this draws on current and appropriate literature in the field. Clarification is also needed to the methodology section, and the interpretation of statistical findings, and thus discussion.

We look forward to receiving your revised manuscript.

Kind regards,

Blanka Golebiowski, PhD BOptom

Academic Editor

PLOS ONE

Journal Requirements:

2. Please include captions for your Supporting Information files at the end of your manuscript, and update any in-text citations to match accordingly. Please see our Supporting Information guidelines for more information: http://journals.plos.org/plosone/s/supporting-information

Reviewers' comments:

Reviewer's Responses to Questions

**Comments to the Author**

1. Is the manuscript technically sound, and do the data support the conclusions?

Reviewer #1: Partly

Reviewer #2: Partly

2. Has the statistical analysis been performed appropriately and rigorously? 

Reviewer #1: I Don't Know

Reviewer #2: I Don't Know

3. Have the authors made all data underlying the findings in their manuscript fully available?

Reviewer #1: Yes

Reviewer #2: No

4. Is the manuscript presented in an intelligible fashion and written in standard English?

Reviewer #1: Yes

Reviewer #2: Yes

5. Review Comments to the Author

Reviewer #1: Thank you for asking me to review this concise and easy to read manuscript. I enjoyed reading it.

Title – I don’t think that this reflects the content of the manuscript. The vergence aftereffects were not statistically significant. You haven’t mentioned that your manuscript looks at the difference between walking and sitting while using the phone.

Abstract (conclusion): Final sentence “To ensure pedestrian safety….while walking”. This does not reflect the content of your manuscript, so I think it should be removed.

Introduction, page 3, lines 70-71. “Indeed, research has revealed…..related to visual quality”. These references refer to people with visual impairment, and is a different issue altogether from the content of the rest of the paragraph (and indeed, the manuscript). If you include these references then I think you should explain what visual quality parameters cause the problem and explain how it relates to the content of the manuscript. Otherwise, I think the sentence (and references) should be removed.

Experimental protocols (page 7), 2nd paragraph. “At each visit, the experiment began with…” The word “with” needs to be included.

Experimental protocols (page 7), 2nd paragraph. Three minutes dark adaptation – it’s not clear why did you did this. Was the experiment conducted in the dark? If someone is reading from a display with the luminance at maximum level, then they would not be dark-adapted anyway. Is 3-minutes long enough to dark adapt someone – probably not? Maybe you need to say that participants were seated in a darkened room for 3 minutes – and then explain why.

Results, page 10. “J0” and “J45”. I can guess what you mean by this, but I think it would be better to explain it to the reader rather than assume that they understand.

Discussion, page 14, last sentence of paragraph 1. “The patterns of near heterophoria change were also significantly different….” On the previous page you said that they were not statistically significant but “tended to show more exo-deviation”. I think you should reword this sentence in the discussion because you cannot say that the heterophoria change was significantly different.

Discussion, page 14, paragraph 2, 1st line. “…no matter smartphone usage…”. The sentence doesn’t make sense. Have some words been omitted?

Discussion, page 14, 15, 16. You discuss the change in downward gaze angle can alter the corneal curvature and cause astigmatism, and on page 16 (2nd paragraph) you mention “working distances, gaze direction, walking gait and dynamic visual environment” as possible stressors for the eyes. However, there is no mention of neck and head angle. I know that you didn’t measure this in this study, but it might have affected your results. For example, some people hold their phone below eye level and look downwards (and this could set up the lid-effects and astigmatism). Other people don’t move their eyes downward (i.e. the eyes are in a straight-ahead gaze position) but they bend/extend their head and neck forward to view the phone in their hands. This will have a different effect on the eye/lid dynamics. I think that you should mention this issue in the discussion.

Discussion, page 14, paragraph 2. You have cited other studies that have shown induced astigmatic changes with reading and microscopy. How does the magnitude of the changes you observed compare with the magnitude of changes reported in the other studies?

Reviewer #2: Thank you for the opportunity to review this paper. Although the writing is in standard English, the logic throughout the manuscript is unclear. With the following suggestions, the manuscript may be transformed into a useful addition to the literature.

Abstract:

Authors should consider the repeatability of heterophoria measurements. There are many papers and vergence after effect measurements in determining if the statistically significant finding for the vergence after effect has any real-life or clinical significance.

Introduction:

Argument in paragraph 2 of the introduction that mobility-related accidents are closely to related to visual quality is not well argued. Authors should consider the effects of divided attention and the smartphone will occupy the central visual field so the user is relying on more peripheral vision to navigate the environment. The authors could look to data on simulated and real central and visual impairment to understand the varied effects of location of visual field defects or scotomas have on mobility performance. There is a large literature on this. Visual stress is likely to be the least important factor when it comes to mobility problems and most patients who experience visual stress will avoid the activity unless it is absolutely necessary (e.g. work and study) rather than video streaming, playing games etc. Although the authors have raised issues with smartphone use, the argument is not logical. Visual stress also has more than one meaning in the vision science literature so this term should be defined within the manuscript.

The term “optical and vergence aftereffects” is not commonly used. It might be that the authors are thinking of short-term adaptation of the fusional vergence systems. The authors should review and include some of that literature in here and revise the language used. The authors should also clearly state the hypothesis, expected findings, in lieu of current understanding of fusional vergence adaptation and how it may be affected by posture and whether they are walking or static.

Methods and Rationale for the study design:

The authors need to provide an explanation of the refractive elements of the eye and their relative contribution to final refractive outcome and explain why the focus is only on the corneal refractive component, and then in the discussion explain what this means for interpretation of the study.

The authors need to explain much more clearly, using equations, the relationship between the magnitude of corneal astigmatism as measured using Zernike polynomials and in Dioptres. It is unclear what the clinical significance of the findings are. It is unclear whether Figures 1D and E are the same data depicted in two different units, or if post-condition refraction was conducted. What would the significance of a corneal shift in astigmatism be without an understanding of how the rest of the ocular elements are changed. For example, the act of convergence may potentially change the relative location of the refractive elements within the eye.

Results:

I found the results difficult to follow and could not determine whether Bonferroni correction was applied correctly. The authors should present all results clearly in a table with the multiple comparisons clearly shown. Non-significant findings after Bonferroni correction should be considered non-significant.

Discussion:

This should be revised after considering the additional literature that the authors will have included in the Introduction.

6. PLOS authors have the option to publish the peer review history of their article (what does this mean?). If published, this will include your full peer review and any attached files.

Reviewer #1: No

Reviewer #2: No

---

## [Author Response · Author response to Decision Letter 0]

13 Jul 2020

Reviewer #1: 

Thank you for asking me to review this concise and easy to read manuscript. I enjoyed reading it.

Title – I don’t think that this reflects the content of the manuscript. The vergence aftereffects were not statistically significant. You haven’t mentioned that your manuscript looks at the difference between walking and sitting while using the phone.

Ans: Thank you for the comment. The title has been changed to “Changes in Corneal Astigmatism and Near Heterophoria after Smartphone Use while Walking and Sitting”. The short title has also been rephrased to “Optical and Heterophoria Effects after Smartphone Use”.

Abstract (conclusion): Final sentence “To ensure pedestrian safety….while walking”. This does not reflect the content of your manuscript, so I think it should be removed.

Ans: We agree with the reviewer's comment, the sentence has been removed.

Introduction, page 3, lines 70-71. “Indeed, research has revealed…..related to visual quality”. These references refer to people with visual impairment, and is a different issue altogether from the content of the rest of the paragraph (and indeed, the manuscript). If you include these references then I think you should explain what visual quality parameters cause the problem and explain how it relates to the content of the manuscript. Otherwise, I think the sentence (and references) should be removed.

Ans: By considering both reviewers’ comments, we agree that this sentence is irrelevant to the current content of the manuscript and it has been removed. 

Experimental protocols (page 7), 2nd paragraph. “At each visit, the experiment began with…” The word “with” needs to be included.

Ans: Thanks. The word “with” is included (P7, L160).

Experimental protocols (page 7), 2nd paragraph. Three minutes dark adaptation – it’s not clear why did you did this. Was the experiment conducted in the dark? If someone is reading from a display with the luminance at maximum level, then they would not be dark-adapted anyway. Is 3-minutes long enough to dark adapt someone – probably not? Maybe you need to say that participants were seated in a darkened room for 3 minutes – and then explain why.

Ans: Sorry for the confusion and thanks for the suggestion. The purpose of “dark adaptation” was to dissipate any transient changes in accommodation and vergence before starting the measurements. We have rewritten the sentence and added relevant references (P7, L158-160).

Results, page 10. “J0” and “J45”. I can guess what you mean by this, but I think it would be better to explain it to the reader rather than assume that they understand.

Ans: Thanks for the comment. We’ve added equations to show the conversion from Zernike coefficients to dioptres and explain J0 and J45 in the Method section (P9, L199-207).

Discussion, page 14, last sentence of paragraph 1. “The patterns of near heterophoria change were also significantly different….” On the previous page you said that they were not statistically significant but “tended to show more exo-deviation”. I think you should reword this sentence in the discussion because you cannot say that the heterophoria change was significantly different.

Ans: We apologize for the mistake. We’ve revised the sentence to emphasize only this trend (P16, L317-318). 

Discussion, page 14, paragraph 2, 1st line. “…no matter smartphone usage…”. The sentence doesn’t make sense. Have some words been omitted?

Ans: Thanks for pointing this out, we believe that the sentence is inappropriate and have deleted it. 

Discussion, page 14, 15, 16. You discuss the change in downward gaze angle can alter the corneal curvature and cause astigmatism, and on page 16 (2nd paragraph) you mention “working distances, gaze direction, walking gait and dynamic visual environment” as possible stressors for the eyes. However, there is no mention of neck and head angle. I know that you didn’t measure this in this study, but it might have affected your results. For example, some people hold their phone below eye level and look downwards (and this could set up the lid-effects and astigmatism). Other people don’t move their eyes downward (i.e. the eyes are in a straight-ahead gaze position) but they bend/extend their head and neck forward to view the phone in their hands. This will have a different effect on the eye/lid dynamics. I think that you should mention this issue in the discussion.

Ans: Thanks for the suggestions. We agree with the reviewer that head and eye movements during the smartphone use might also affect our results. We’ve included this potential factor in the Discussion (P18-19, L384-389). 

Discussion, page 14, paragraph 2. You have cited other studies that have shown induced astigmatic changes with reading and microscopy. How does the magnitude of the changes you observed compare with the magnitude of changes reported in the other studies?

Ans: We’ve added a sentence to compare the magnitude of change in corneal aberration between our study and other studies involving near works (P17, L344-348).

Reviewer #2: 

Thank you for the opportunity to review this paper. Although the writing is in standard English, the logic throughout the manuscript is unclear. With the following suggestions, the manuscript may be transformed into a useful addition to the literature.

Abstract:

Authors should consider the repeatability of heterophoria measurements. There are many papers and vergence after effect measurements in determining if the statistically significant finding for the vergence after effect has any real-life or clinical significance.

Ans: Thanks for your comment. We’ve added a sentence in the Discussion to alert the readers that while the difference in vergence adaptation between walking and sitting was statistically significant, it did not reach clinical significance (P18 L374-376).

Introduction:

Argument in paragraph 2 of the introduction that mobility-related accidents are closely to related to visual quality is not well argued. Authors should consider the effects of divided attention and the smartphone will occupy the central visual field so the user is relying on more peripheral vision to navigate the environment. The authors could look to data on simulated and real central and visual impairment to understand the varied effects of location of visual field defects or scotomas have on mobility performance. There is a large literature on this. Visual stress is likely to be the least important factor when it comes to mobility problems and most patients who experience visual stress will avoid the activity unless it is absolutely necessary (e.g. work and study) rather than video streaming, playing games etc. Although the authors have raised issues with smartphone use, the argument is not logical. Visual stress also has more than one meaning in the vision science literature so this term should be defined within the manuscript.

Ans: Thanks for your comment. We agree with the reviewer that visual stress is not an essential factor in mobility compared to other factors, such as visual attention, scotomas, and visual field deficits. By considering both reviewers’ comments, we have removed the sentence “Indeed, research has revealed…..related to visual quality” (P3, L72).

The term “optical and vergence aftereffects” is not commonly used. It might be that the authors are thinking of short-term adaptation of the fusional vergence systems. The authors should review and include some of that literature in here and revise the language used. 

Ans: Thanks for pointing this out. We’ve revised the terminology and changed “vergence aftereffect” to “vergence adaptation” throughout the manuscript.

The authors should also clearly state the hypothesis, expected findings, in lieu of current understanding of fusional vergence adaptation and how it may be affected by posture and whether they are walking or static.

Ans: Thank you. We’ve added a paragraph in the Introduction to state the rationale and hypothesis of this study (P4-5, L85-106).

Methods and Rationale for the study design:

The authors need to provide an explanation of the refractive elements of the eye and their relative contribution to final refractive outcome and explain why the focus is only on the corneal refractive component, and then in the discussion explain what this means for interpretation of the study.

Ans: Thanks for the comment. We’ve added an explanation of why we focused on corneal aberration and how it might contribute to the variation of the overall optics of the eye in the Method section (P9, L193-197). We’ve also discussed the potential visual consequence of the aberration change in the Discussion (P17, L339-344). 

The authors need to explain much more clearly, using equations, the relationship between the magnitude of corneal astigmatism as measured using Zernike polynomials and in Dioptres. 

Ans: Thanks. We’ve added the equations for the conversion from Zernike coefficients to dioptres in the Method (P9, L199-207).

It is unclear what the clinical significance of the findings are. 

Ans: We’ve added a sentence to discuss the potential visual consequence of the aberration change in the Discussion (P17, L342-344). 

It is unclear whether Figures 1D and E are the same data depicted in two different units, or if post-condition refraction was conducted. 

Ans: Figure 1D and E represent oblique astigmatism with the same unit as the H/V astigmatism (B & C). To make it clearer, we’ve revised the figure legend (P13, L256-260). Sorry for the confusion. 

What would the significance of a corneal shift in astigmatism be without an understanding of how the rest of the ocular elements are changed. For example, the act of convergence may potentially change the relative location of the refractive elements within the eye.

Ans: Due to the restriction of our study design, we could only selectively measure part of the ocular biometric changes. However, we agree with the reviewer that other potential factors could contribute to the variation of the corneal shape and aberrations. We’ve included these factors in the discussion (P16, L331-334), as well as rationale for why we focused on corneal aberration (P9, L193-197).

Results:

I found the results difficult to follow and could not determine whether Bonferroni correction was applied correctly. The authors should present all results clearly in a table with the multiple comparisons clearly shown. Non-significant findings after Bonferroni correction should be considered non-significant.

Ans: Sorry for the confusion. We have added a table to summarize the statistics (P12, L236-242).

Discussion:

This should be revised after considering the additional literature that the authors will have included in the Introduction.

Ans: The Discussion section is revised. Thanks.

---

## [Decision Letter · Decision Letter 1]

23 Oct 2020

PONE-D-20-16428R1

Changes in corneal astigmatism and near heterophoria after smartphone use while walking and sitting

PLOS ONE

Dear Dr. Leung,

Thank you for submitting your manuscript to PLOS ONE. After careful consideration, we feel that it has merit but does not fully meet PLOS ONE’s publication criteria as it currently stands. Therefore, we invite you to submit a revised version of the manuscript that addresses the points raised during the review process.

We look forward to receiving your revised manuscript.

Kind regards,

Blanka Golebiowski, PhD BOptom

Academic Editor

PLOS ONE

Reviewers' comments:

Reviewer's Responses to Questions

**Comments to the Author**

1. If the authors have adequately addressed your comments raised in a previous round of review and you feel that this manuscript is now acceptable for publication, you may indicate that here to bypass the “Comments to the Author” section, enter your conflict of interest statement in the “Confidential to Editor” section, and submit your "Accept" recommendation.

Reviewer #1: All comments have been addressed

Reviewer #2: (No Response)

2. Is the manuscript technically sound, and do the data support the conclusions?

Reviewer #1: Yes

Reviewer #2: Yes

3. Has the statistical analysis been performed appropriately and rigorously? 

Reviewer #1: I Don't Know

Reviewer #2: Yes

4. Have the authors made all data underlying the findings in their manuscript fully available?

Reviewer #1: Yes

Reviewer #2: Yes

5. Is the manuscript presented in an intelligible fashion and written in standard English?

Reviewer #1: Yes

Reviewer #2: Yes

6. Review Comments to the Author

Reviewer #1: Well done. You've done a great job revising this manuscript. I especially like the inclusion of Table 2.

Reviewer #2: Thank you for the revision. Only one additional query has come up. When watching the movie on the smart phone, were they watching a movie in a language that required them to read the subtitles? In other words, were the subtitles an essential part of the experimental condition?

7. PLOS authors have the option to publish the peer review history of their article (what does this mean?). If published, this will include your full peer review and any attached files.

Reviewer #1: No

Reviewer #2: No

---

## [Author Response · Author response to Decision Letter 1]

31 Oct 2020

Reviewer #1: 

Well done. You've done a great job revising this manuscript. I especially like the inclusion of Table 2.

Thank you very much for reviewing this manuscript and providing valuable comments.

Reviewer #2: 

Thank you for the revision. Only one additional query has come up. When watching the movie on the smart phone, were they watching a movie in a language that required them to read the subtitles? In other words, were the subtitles an essential part of the experimental condition?

Thank you very much! It’s an excellent question. Sorry that we have missed it in the previous version. We played a Korean variety show called “Running Man”, a popular foreign TV program in Hong Kong, for 30 minutes with Chinese subtitle (subtitle’s size: ~ 3mm). None of the participants knew Korean, and reading the subtitle was necessary. This information has been added in the revised version (P7, L159-161).

---

## [Editor Report · Decision Letter 2]

16 Nov 2020

Changes in corneal astigmatism and near heterophoria after smartphone use while walking and sitting

PONE-D-20-16428R2

Dear Dr. Leung,

We’re pleased to inform you that your manuscript has been judged scientifically suitable for publication and will be formally accepted for publication once it meets all outstanding technical requirements.

Kind regards,

Blanka Golebiowski, PhD BOptom

Academic Editor

PLOS ONE
---

## [Editor Report · Acceptance letter]

19 Nov 2020

PONE-D-20-16428R2 

Changes in corneal astigmatism and near heterophoria after smartphone use while walking and sitting 

Dear Dr. Leung:

I'm pleased to inform you that your manuscript has been deemed suitable for publication in PLOS ONE. Congratulations! Your manuscript is now with our production department. 

Kind regards, 

on behalf of

Associate Professor Blanka Golebiowski 

Academic Editor

PLOS ONE